# RECOVERING GEOMETRIC INFORMATION WITH LEARNED TEXTURE PERTURBATIONS

## ABSTRACT

Regularization is used to avoid overfitting when training a neural network; unfortunately, this reduces the attainable level of detail hindering the ability to capture high-frequency information present in the training data. Even though various approaches may be used to re-introduce high-frequency detail, it typically does not match the training data and is often not time coherent. In the case of network inferred cloth, these sentiments manifest themselves via either a lack of detailed wrinkles or unnaturally appearing and/or time incoherent surrogate wrinkles. Thus, we propose a general strategy whereby high-frequency information is procedurally embedded into low-frequency data so that when the latter is smeared out by the network the former still retains its high-frequency detail. We illustrate this approach by learning texture coordinates which when smeared do not in turn smear out the high-frequency detail in the texture itself but merely smoothly distort it. Notably, we prescribe perturbed texture coordinates that are subsequently used to correct the over-smoothed appearance of inferred cloth, and correcting the appearance from multiple camera views naturally recovers lost geometric information.

## 1 INTRODUCTION

Since neural networks are trained to generalize to unseen data, regularization is important for reducing overfitting, see e.g. Goodfellow et al. (2016); Scholkopf & Smola (2001). However, regularization also removes some of the high variance characteristic of much of the physical world. Even though high-quality ground truth data can be collected or generated to reflect the desired complexity of the outputs, regularization will inevitably smooth network predictions. Rather than attempting to directly infer high-frequency features, we alternatively propose to learn a low-frequency space in which such features can be embedded.

We focus on the specific task of adding high-frequency wrinkles to virtual clothing, noting that the idea of learning a low-frequency embedding may be generalized to other tasks. Because cloth wrinkles/folds are high-frequency features,

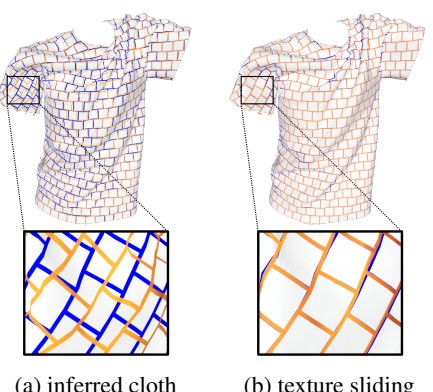

(a) inferred cloth      (b) texture sliding

Figure 1: Texture coordinate perturbations (texture sliding) reduce shape inference errors: ground truth (blue), prediction (orange).

existing deep neural networks (DNNs) trained to infer cloth shape tend to predict overly smooth meshes Alldieck et al. (2019a); Daněřek et al. (2017); Guan et al. (2012); Gundogdu et al. (2019); Jin et al. (2020); Lahner et al. (2018); Natsume et al. (2019); Santesteban et al. (2019); Wang et al. (2018); Patel et al. (2020). Rather than attempting to amend such errors directly, we perturb texture so that the rendered cloth mesh *appears* to more closely match the ground truth. See Figure 1. Then given texture perturbations from at least two unique camera views, 3D geometry can be accurately reconstructed Hartley & Sturm (1997) to recover high-frequency wrinkles. Similarly, for AR/VR applications, correcting visual appearance from two views (one for each eye) is enough to allow the viewer to accurately discern 3D geometry. Our proposed texture coordinate perturbations are highly

dependent on the camera view. Thus, we demonstrate that one can train a separate texture sliding neural network (TSNN) for each of a finite number of cameras laid out into an array and use nearby networks to interpolate results valid for any view enveloped by the array. Although an approach similar in spirit might be pursued for various lighting conditions, this limitation is left as future work since there are a great deal of applications where the light is ambient/diffuse/non-directional/etc. In such situations, this further complication may be ignored without significant repercussion.

## 2  RELATED WORK

**Cloth:** While physically-based cloth simulation has matured as a field over the last few decades Baraff & Witkin (1998); Baraff et al. (2003); Bridson et al. (2002; 2003); Selle et al. (2008), data-driven methods are attractive for many applications. There is a rich body of work in reconstructing cloth from multiple views or 3D scans, see e.g. Bradley et al. (2008b); Franco et al. (2006); Vlasic et al. (2008). More recently, optimization-based methods have been used to generate higher resolution reconstructions Huang et al. (2015); Pons-Moll et al. (2017); Wu et al. (2012); Yang et al. (2016). Some of the most interesting work focuses on reconstructing the body and cloth separately Bălan & Black (2008); Neophytou & Hilton (2014); Yang et al. (2018); Zhang et al. (2017). With advances in deep learning, one can aim to reconstruct 3D cloth meshes from single views. A number of approaches reconstruct a joint cloth/body mesh from a single RGB image Alldieck et al. (2019a;b); Natsume et al. (2019); Onizuka et al. (2020); Saito et al. (2019; 2020), RGB-D image Yu et al. (2019), or video Alldieck et al. (2018a;b); Habermann et al. (2019); Xu et al. (2018). To reduce the dimensionality of the output space, DNNs are often trained to predict the pose/shape parameters of human body models such as SCAPE Anguelov et al. (2005) or SMPL Loper et al. (2015) (see also Pavlakos et al. (2019)). Habermann et al. (2019); Natsume et al. (2019); Varol et al. (2018) leverage predicted pose information to infer shape. When only the garment shape is predicted, a number of recent works output predictions in UV space to represent geometric information as pixels Daněřek et al. (2017); Jin et al. (2020); Lahner et al. (2018), although others Gundogdu et al. (2019); Santesteban et al. (2019); Patel et al. (2020) define loss functions directly in terms of the 3D cloth vertices.

**Wrinkles and Folds:** Cloth realism can be improved by introducing wrinkles and folds. In the graphics community, researchers have explored both procedural and data-driven methods for generating wrinkles De Aguiar et al. (2010); Guan et al. (2012); Hahn et al. (2014); Müller & Chentanez (2010); Rohmer et al. (2010); Wang et al. (2010). Other works add real-world wrinkles as a postprocessing step to improve smooth captured cloth: Popa et al. (2009) extracts the edges of cloth folds and then applies space-time deformations, Robertini et al. (2014) solves for shape deformations directly by optimizing over all frames of a video sequence. Recently, Lahner et al. (2018) used a conditional Generative Adversarial Network Mirza & Osindero (2014) to generate normal maps as proxies for wrinkles on captured cloth.

**Geometry:** More broadly, deep learning on 3D meshes falls under the umbrella of *geometric deep learning*, which was coined by Bronstein et al. (2017) to characterize learning in non-Euclidean domains. Scarselli et al. (2008) was one of the earliest works in this area and introduced the notion of a Graph Neural Network (GNN) in relation to CNNs. Subsequent works similarly extend the CNN architecture to graphs and manifolds Boscaini et al. (2016); Maron et al. (2017); Masci et al. (2015); Monti et al. (2017). Kostrikov et al. (2018) introduces a latent representation that explicitly incorporates the Dirac operator to detect principal curvature directions. Tan et al. (2018) trains a mesh generative model to generate novel meshes outside an original dataset. Returning to the specific application of virtual cloth, Jin et al. (2020) embeds a non-Euclidean cloth mesh into a Euclidean pixel space, making it possible to directly use CNNs to make non-Euclidean predictions.

## 3  METHODS

We define *texture sliding* as the changing of texture coordinates on a per-camera basis such that any point which is visible from some stereo pair of cameras can be triangulated back to its ground truth position. Other stereo reconstruction techniques can also be used in place of triangulation because the *images* we generate are consistent with the ground truth geometry. See e.g. Bradley et al. (2008a); Hartley & Sturm (1997); Seitz et al. (2006).

## 3.1 PER-VERTEX DISCRETIZATION

Since the cloth mesh is discretized into vertices and triangles, we take a per-vertex, not a per-point, approach to texture sliding. Our proposed method (see Section 4.1) computes per-vertex texture coordinates on the inferred cloth that match those of the ground truth as seen by the camera under consideration. Then during 3D reconstruction, barycentric interpolation is used to find the subtriangle locations of the texture coordinates corresponding to ground truth cloth vertices. This assumes linearity, which is only valid when the triangles are small enough to capture the inherent nonlinearities in a piecewise linear sense; moreover, folds and wrinkles can create significant nonlinearity. See Figure 2.

## 3.2 OCCLUSION BOUNDARIES

Accurate 3D reconstruction requires that a vertex of the ground truth mesh be visible from at least two cameras *and* that camera projections of the vertex to the inferred cloth exist and are valid. However, occlusions can derail these assumptions.

First, consider things from the standpoint of the inferred cloth. For a given camera view, some inferred cloth triangles will not contain any visible pixels, and we denote a vertex as occluded when none of its incident triangles contain any visible pixels. Although we do not assign perturbed texture coordinates to occluded vertices (i.e. they keep their original texture coordinates, or a perturbation of zero), we do aim to keep the texture coordinate perturbation function smooth (see Section 4.2). In addition, there will be so called non-occluded vertices in the inferred cloth that do not project to visible pixels of the ground truth cloth. This often occurs near silhouette boundaries where the inferred cloth silhouette is sometimes wider than the ground truth cloth silhouette. These vertices are also treated as occluded, similar to those around the back side of the cloth behind the silhouette, essentially treating some extra vertices near occlusion boundaries as also being occluded. See Figure 3a.

Next, consider things from the standpoint of the ground truth cloth. For example, consider the case where all the cameras are in the front, and vertices on the back side of the ground truth cloth are not visible from any camera. The best one can do in reconstructing these occluded vertices is to use the inferred cloth vertex positions; however, care should be taken near occlusion boundaries to smoothly taper between our texture sliding 3D reconstruction and the inferred cloth prediction. A simple approach is to extrapolate/smooth the geometric difference between our texture sliding 3D reconstruction and the inferred cloth prediction to occluded regions of the mesh. Once again, the definition of occluded vertices needs to be broadened for silhouette consideration. Not only will vertices not visible from at least two cameras have to be considered occluded, but vertices that don't project to the interior of an inferred cloth triangle with *valid* texture coordinate perturbations will also have to be considered occluded. See Figure 3b.

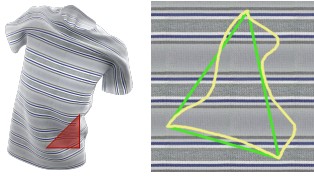 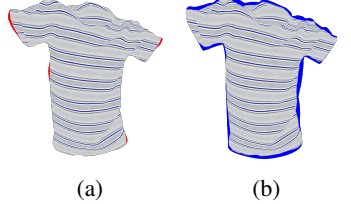

(a)       (b)

Figure 2: Consider an extreme case, where the inferred cloth has a quite large triangle (shown in red). That triangle should encompass the nonlinear texture region outlined in yellow (shown in pattern space). Note: the yellow curve was generated by sampling the ground truth cloth's texture coordinates along the projected edges of the red triangle. The linearity assumption implied by barycentric interpolation instead uses the region outlined in green.

Figure 3: The method discussed in Section 4.1 can fail near silhouettes of the inferred and ground truth cloth meshes, in which case smoothness assumptions are used (see Section 4.2). In (a), inferred triangles with at least one vertex falling outside the silhouette of the ground truth mesh are colored red. In (b), ground truth triangles with at least one vertex falling outside the silhouette of the inferred mesh are colored blue.

## 4 DATASET GENERATION

Let $C = \{X, T\}$ be a cloth triangulated surface with $n$ vertices $X \in \mathbb{R}^{3n}$ and texture coordinates $T \in \mathbb{R}^{2n}$. We assume that mesh connectivity remains fixed throughout. The ground truth cloth mesh $C_G(\theta) = \{X_G(\theta), T_G\}$ depends on the pose $\theta$. Given a pre-trained DNN (we use the network from Jin et al. (2020)), the inferred cloth $C_N(\theta) = \{X_N(\theta), T_G\}$ is also a function of the pose $\theta$. Our objective is to replace the ground truth texture coordinates $T_G$ with perturbed texture coordinates $T_N(\theta, v)$, i.e. to compute $C_N'(\theta, v) = \{X_N(\theta), T_N(\theta, v)\}$ where $T_N(\theta, v)$ depends on both the pose $\theta$ and the view $v$. Even though $T_N(\theta, v)$ is in principle valid for all $v$ using interpolation (see Section 6.3), training data $T_N(\theta, v_p)$ is only required for a finite number of camera views $v_p$. For each camera $p$, we also only require training data for finite number of poses $\theta_k$, i.e. we require $T_N(\theta_k, v_p)$, which is computed from $T_G$ using $X_G(\theta_k)$, $X_N(\theta_k)$, and $v_p$.

### 4.1 TEXTURE COORDINATE PROJECTION

We project texture coordinates to the inferred cloth vertices $X_N(\theta_k)$ from the ground truth cloth mesh $C_G(\theta_k)$ using ray intersection. For each inferred cloth vertex in $X_N(\theta_k)$, we cast a ray from camera $p$'s aperture through the vertex and find the first intersection with the ground truth mesh $C_G(\theta_k)$; subsequently, $T_G$ is barycentrically interpolated to the point of intersection and assigned to the inferred cloth vertex as its $T_N(\theta_k, v_p)$ value. See Figure 4. Rays are only cast for inferred cloth vertices that have at least one incident triangle with a nonzero area subregion visible to camera $p$. Also, a ground truth texture coordinate value is only assigned to an inferred cloth vertex when the point of intersection with the ground truth mesh is visible to camera $p$. We store and learn texture coordinate displacements $d_{v_p}(\theta_k) = T_N(\theta_k, v_p) - T_G$. Af-

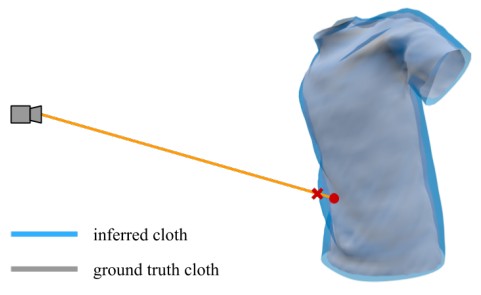

Figure 4: Illustration of the ray intersection method for transferring texture coordinates to the inferred cloth from the ground truth cloth. Texture coordinates for the inferred cloth vertex (red cross) are interpolated from the ground truth mesh to the point of ray intersection (red circle).

ter this procedure, any remaining vertices of the inferred cloth that have not been assigned $d_{v_p}(\theta_k)$ values are treated as occluded and handled via smoothness considerations as discussed in Section 4.2.

### 4.2 OCCLUSION HANDLING

Some vertices of the inferred cloth mesh remain unassigned with $d_{v_p}(\theta_k) = 0$ after executing the algorithm outlined in Section 4.1. This creates a discontinuity in $d_{v_p}(\theta_k)$ which excites high frequencies that require a more complex network architecture to capture. In order to alleviate demands on the network, we smooth $d_{v_p}(\theta_k)$ as follows. First, we use the Fast Marching Method on triangulated surfaces Kimmel & Sethian (1998) to generate a signed distance field. Then, we extrapolate $d_{v_p}(\theta_k)$ normal to the distance field into the unassigned region, see e.g. Osher & Fedkiw (2002). Finally, a bit of averaging is used to provide smoothness, while keeping the assigned values of $d_{v_p}(\theta_k)$ unchanged. Alternatively, one could solve a Poisson equation as in Cong et al. (2015) while using the assigned $d_{v_p}(\theta_k)$ as Dirichlet boundary conditions.

## 5 NETWORK ARCHITECTURE

A separate texture sliding neural network (TSNN) is trained for each camera $p$; thus, we drop the $v_p$ notation in this section. The loss is defined over all poses $\theta_k$ in the training set

$$\mathcal{L} = \sum_{\theta_k} \left\| d(\theta_k) - \hat{d}(\theta_k) \right\|_2 \tag{1}$$

to minimize the difference between the desired displacements $d(\theta_k)$ and predicted displacements $\hat{d}(\theta_k)$. The inferred cloth data we chose to correct are predictions of the T-shirt meshes from Jin

et al. (2020), each of which contains about 3,000 vertices. The dataset spans about 10,000 different poses generated from a scanned garment using physically-based simulation, and includes texture coordinates for the garment mesh. To improve the resolution, we up-sampled each cloth mesh by subdividing each triangle into four subtriangles. Notably, our texture sliding approach can be used to augment the results of any dataset for which ground truth and inferred training examples are available. Moreover, it is trivial to increase the resolution of any such dataset simply by subdividing triangles. Note that perturbations of the subdivided geometry are unnecessary, as we merely desire more sample points (to address Figure 2). Finally, we applied an 80-10-10 training-validation-test set split.

Similar to Jin et al. (2020), the displacements $d(\theta_k)$ are stored as pixel-based cloth images for the front and back sides of the T-shirt, though we still output per-vertex texture coordinate displacements in UV space. See Figure 5 for an overview of the network architecture. Given input joint transformation matrices of shape $1 \times 1 \times 90$, TSNN applies a series of transpose convolution, batch normalization, and ReLU activation layers to upsample the input to $512 \times 512 \times 4$. The first two dimensions of the output tensor represent the predicted displacements for the front side of the T-shirt, and the remaining two dimensions represent those for the back side.

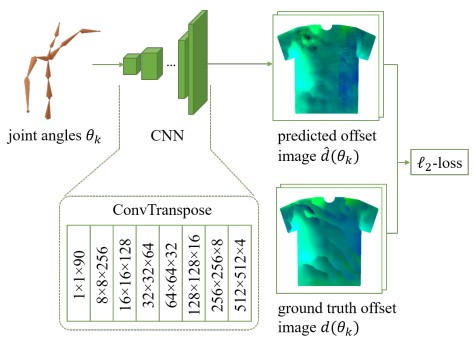

Figure 5: Texture sliding neural network (TSNN) architecture.

## 6 EXPERIMENTS

In Section 6.1, we quantify the data generation approach of Section 4 and highlight the advantages of mesh subdivision for up-sampling. In Section 6.2, we evaluate the predictions made by our trained texture sliding neural network (TSNN). In Section 6.3, we demonstrate the interpolation of texture sliding results to novel views between a finite number of cameras. Finally, in Section 6.4, we use multi-view texture sliding to reconstruct 3D geometry.

### 6.1 DATASET GENERATION AND EVALUATION

We aim to have the material coordinates of the cloth be in the correct locations as viewed by multiple cameras, so that the material can be accurately 3D reconstructed with point-wise accuracy. As such, our error metric is a bit more stringent than that commonly used because our aim is to reproduce the actual material behavior, not merely to mimic its look (e.g. , by perturbing normal vectors to create shading consistent with wrinkles in spite of the cloth being smooth, as in Lahner et al. (2018)). In order to elucidate this, consider a two-step approach where one first approximates a smooth cloth mesh and then perturbs that mesh to add wrinkles (similar to Santesteban et al. (2019)). In order to preserve area and achieve the correct material behavior, material in the vicinity of a newly forming wrinkle should slide laterally towards that wrinkle as it is formed. Merely non-physically stretching the material in order to create a wrinkle may look plausible, but does not admit the correct material behavior. In fact, the texture would be unrealistically stretched as well, although this is less apparent visually when using simple textures.

Since texture coordinates provide a proxy surface parameterization for material coordinates, we measure texture coordinate errors in a per-pixel fashion comparing between the ground truth and inferred cloth at the center of each pixel. Figure 6a shows results typical for cloth inferred using the network from Jin et al. (2020), and Figure 6b shows the highly improved results obtained on the same inferred geometry using our texture sliding approach (with 1 level of subdivision). Note that the vast majority of the errors in Figure 6b occur near the wrinkles where the nonlinearities illustrated in Figure 2 are most prevalent. In Figure 6c, we deform the vertices of the inferred cloth mesh so that they lie exactly on the ground truth mesh in order to mimic a two-step approach (as discussed above). Note how our error metric captures the still rather large errors in the material coordinates (and thus cloth vertex positions) in spite of the mesh in Figure 6c appearing to have the same wrinkles and folds as the ground truth mesh. Figure 7 compares the local compression and extension energies of

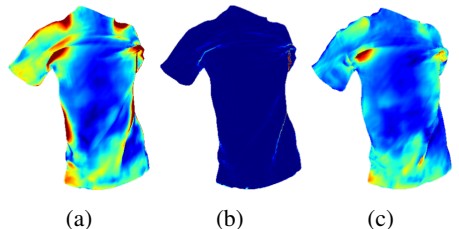
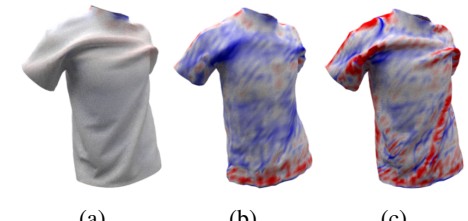

| (a) | (b) | (c) | | (a) | (b) | (c) |

Figure 6: Per-pixel texture coordinate errors before (a) and after (b) applying texture sliding to the inferred cloth output by the network of Jin et al. (2020). The result of a two-step process (c) may well match the ground truth in a visual sense, whilst still having quite large errors in material coordinates. Blue = 0, red ≥ 0.04.

Figure 7: Local compression (blue) and extension (red) energies for a sample pose, comparing the ground truth cloth (a), the inferred cloth (b), and the result of a two-step process (c). In spite of the cloth mesh in (c) bearing visual resemblance to the ground truth in (a), it still has quite erroneous deformation energies.

the ground truth mesh (Figure 7a), the inferred cloth mesh (Figure 7b), and the result of this two-step process (Figure 7c). In spite of the untextured mesh in Figure 7c bearing visual similarity to the ground truth in Figure 7a, it still has rather large errors in deformation energy.

Figure 8 illustrates the efficacy of subdividing the cloth mesh to get more samples for texture sliding. The particular ground truth cloth wrinkle shown in Figure 8e is not captured by the inferred cloth geometry shown in Figure 8a. The texture sliding result shown in Figure 8b better represents the ground truth cloth. Figures 8c and 8d show how subdividing the inferred cloth mesh one and two times (respectively) progressively alleviates errors emanating from the linearity assumption illustrated in Figure 2. Table 1 shows quantitative results comparing the inferred cloth to texture sliding with and without subdivision.

| Method | SqrtMSE ($\times 10^{-3}$) |
|---|---|
| Jin et al. (2020) | $24.496 \pm 6.9536$ |
| TS | $5.2662 \pm 2.2320$ |
| TS + subdivision | $3.5645 \pm 1.6617$ |

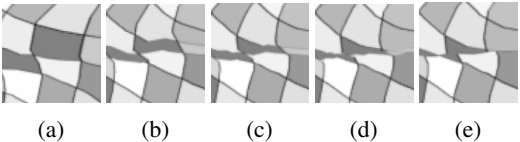

| (a) | (b) | (c) | (d) | (e) |

Table 1: Per-pixel square root of mean squared error (SqrtMSE) for the entire dataset.

Figure 8: As the inferred cloth mesh (a) is subdivided, texture sliding (b-d) moves the inferred mesh's appearance closer to the ground truth (e).

## 6.2 NETWORK TRAINING AND INFERENCE

The network was trained using the Adam optimizer Kingma & Ba (2014) with a $10^{-3}$ learning rate in PyTorch Paszke et al. (2017). As mentioned earlier, we subdivided the mesh triangles once. Figure 9 shows a typical prediction on a test set example, including the per-pixel errors in predicted texture coordinates. While the TSNN is able to recover the majority of the shirt, it struggles near wrinkles. Figure 10 highlights a particular wrinkle comparing the inferred cloth (Figure 10a) and the results of the TSNN before (Figure 10b) and after (Figure 10c) subdivision to the ground truth (Figure 10d). Table 2 shows quantitative results comparing the inferred cloth to TSNN results with and without subdivision.

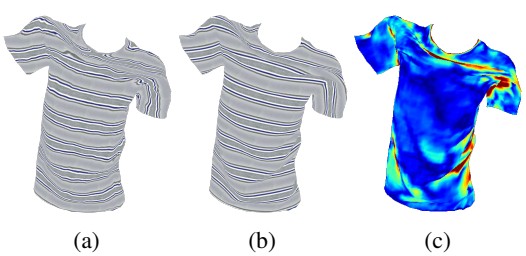

| (a) | (b) | (c) |

Figure 9: A typical test set example prediction. (a) $\hat{C}'_N$. (b) $C'_N$. (c) Per-pixel errors (blue = 0, red ≥ 0.04).

| Network | SqrtMSE ($\times 10^{-3}$) |
|---|---|
| Jin et al. (2020) | $24.871 \pm 7.0613$ |
| TSNN | $13.335 \pm 4.2924$ |
| TSNN + subdivision | $13.591 \pm 4.5194$ |

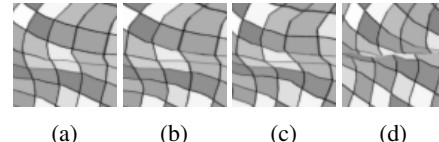

(a)     (b)     (c)     (d)

Table 2: Per-pixel SqrtMSE for the test set. In-spite of Table 1 demonstrating that subdivision improves the ground truth TS data, the improvements are not uniformly realized by the TSNN (which we discuss in the supplemental material).

Figure 10: The results of the TSNN before (b) and after (c) subdivision, as compared to the ground truth (d). In spite of Table 2, some wrinkles are better resolved by the TSNN after subdivision. The inferred mesh with ground truth texture coordinates is shown in (a).

## 6.3 INTERPOLATING TO NOVEL VIEWS

Given a finite number of camera views $v_p$, one can specify a new view enveloped by the array using a variety of interpolation methods. For the sake of demonstration, we take a simple approach assuming that one can interpolate via $v = \sum_p w_p v_p$, and then use these same weights to compute $T_N(\theta_k, v) = \sum_p w_p T_N(\theta_k, v_p)$. This same equation is also used for $\hat{T}_N(\theta_k, v)$. Figure 11 shows the results obtained by linearly interpolating between two camera views. Note how the largest errors appear near areas occluded by wrinkles, where one (or both) of the cameras has no valid texture sliding results and instead uses the inferred cloth textures. This can be alleviated by using more cameras placed closer together.

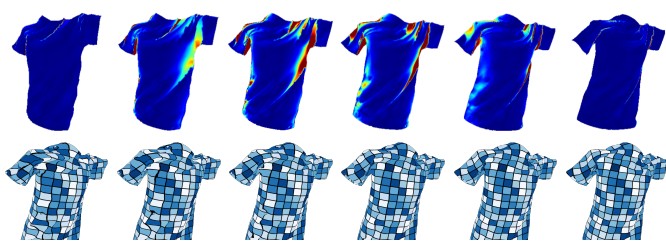

Figure 11: Given two camera views (far left and far right images), texture sliding can be linearly interpolated to novel views between them. The top row shows per-pixel errors (blue = 0, red $\geq 0.04$), and the bottom row shows the cloth from a *fixed* front-facing view to illustrate how the interpolated texture changes as a function of the chosen novel view.

## 6.4 3D RECONSTRUCTION

In order to reconstruct the 3D position of a vertex of the ground truth mesh, we take the usual approach of finding rays that pass through that vertex and the camera aperture for a number of cameras. Then given at least two rays, one can triangulate a 3D point that is minimal distance from all the rays. We can do this without solving the typical image to image correspondence problem because we know the ground truth texture coordinates for any given vertex. Thus, we merely have to find the ray that passes through the camera aperture and the ground truth texture coordinate for the vertex under consideration.

To find a ground truth texture coordinate on a texture corrected inferred cloth mesh $C'_N(\theta_k, v)$, or $\hat{C}'_N(\theta_k, v)$, we first find the triangle containing that texture coordinate. This can be done quickly by using a hierarchical bounding box structure where the base level boxes around each triangle are defined using the min/max texture coordinates at the three vertices. Then one can write the barycentric interpolation formula that interpolates the triangle vertex texture coordinates to obtain the given ground truth texture coordinate, and subsequently invert the matrix to solve for the weights. These weights determine the sub-triangle position of the vertex under consideration (taking care to note that different answers are obtained in 3D space versus screen space, since the camera projection is nonlinear). Figure 12 shows the 3D reconstruction of a test set example using texture sliding (Figure 12c) and the TSNN (Figure 12d). To remove reconstruction noise generated by network inference

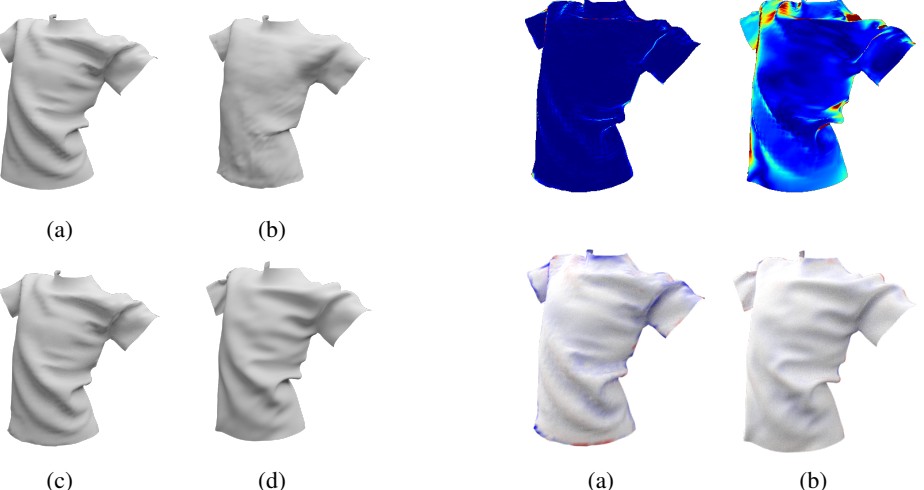

Figure 12: The ground truth (a) and inferred cloth (b) compared to the 3D reconstructions obtained using texture sliding (c) and the TSNN (d).

Figure 13: Per-pixel errors (top) and local compression/extension energies (bottom) for Figure 12c (a) and Figure 12d (b).

errors in Figure 12d, we used the postprocess from Geng et al. (2020); although, there are many other smoothing options in the literature that one might also consider. Figure 13 compares the per-pixel errors and local compression/extension energies of Figures 12c and 12d.

## 7 DISCUSSION AND FUTURE WORK

There are many disparate applications for clothing including for example video games, AR/VR, Hollywood special effects, virtual try-on and shopping, scene acquisition and understanding, and even bullet proof vests and soft armor. Various scenarios define accuracy or fidelity in vastly different ways. So while it is typical to state that one cares about more than just the visual appearance (or "graphics"), often those aiming for predictive capability still make concessions. For example, wherein Santesteban et al. (2019) proposes a network that well predicts wrinkles mapped to new body types, the discussion in Lahner et al. (2018) implies that the horizontal wrinkles predicted by Santesteban et al. (2019) are more characteristic of inaccurate physical simulation than real-world behavior. Instead, Lahner et al. (2018) strives for more vertical wrinkles to better match their data, but they accomplish this by predicting lighting to match an image while accepting overly smooth geometry. And as we have shown in Figure 7c, predicting the correct geometry still allows for rather large errors in the deformation (see Geng et al. (2020)).

In light of this, we state the problem of most interest to us: Our aim is to study the efficacy of using deep neural networks to aid in the modeling of material behavior, especially for those materials for which predictive methods do not currently exist because of various unknowns including friction, material parameters (for cloth and body), etc. Given this goal, we focus on the accurate prediction of material coordinates, which are a super set of deformation, geometry, lighting, visual plausibility, etc.

As demonstrated by the remarkably accurate 3D reconstruction in Figure 12c (see 13a), our approach to encoding high frequency wrinkles into lower frequency texture coordinates (i.e. texture sliding) works quite well. It can be used as a post-process to any existing neural network to capture lost details (as long as ground truth and inferred training examples are available); moreover, we showed that trivial subdivision could be used to increase the sampling resolution to limit linearization artifacts. One needs to take care when training the texture sliding neural network (TSNN) since inference errors can cause reconstruction noise. Thus, as future work, we plan on experimenting with the network architecture, the size of the image used in the CNN, the smoothing methods near occlusion boundaries, the amount of subdivision, etc. In addition, it would be interesting to consider more savvy multiview 3D reconstruction methods (particularly ones that employ DNNs; then, one might train the whole process end-to-end).

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
