# OpenReview forum: "Recovering Geometric Information with Learned Texture Perturbations"
_ICLR.cc/2021/Conference — Reject_

### Official Review · AnonReviewer1 · 2020-10-25
**An ok paper, that could be strengthened**

**Rating:** 4
**Confidence:** 3

**Review:**

The paper proposes a method to correct high-frequencies details in the textures of animated clothes. The main idea is to train a network to learn the 2D offset in the UV space for a given pose and view. The paper shows results using a t-shirt, on interpolating to novel views and 3D reconstruction by correcting the output of a state of the art method.

PROS
1) Comparison:
The paper improves a state-of-the-art method (Jin et al. 2020), suggesting that the proposed module could be applied by other methods as post-processing.

2) Applicability:
I think the idea could be easily applicable in other tissues domains (e.g. soft-tissue modeling for humans, animals fur).

CONS
1) Method and Experiments:

- I think the proposed methodology is not particularly innovative, and it acts in a controlled scenario: the method is a CNN developed to work in a (virtual) setup that requires several prior knowledge: the view (also, it need to be trained for every different one), the pose parameters (making it also dependent on the animation system), the garment, the texture, and the light condition. Finally, the shown examples and experiments are limited to a few setups, without stressing the limits of the method.
I am wondering how does the method performs training a different kind of garments with higher mobility (e.g. skirts) or subject to non-isometric deformations (e.g. zip clothes) or with irregular textures at higher frequencies (e.g. a shirt with a clear logo).
- I am not surprised that the method improves the starting one, and I would suggest to compare it with others refinement baselines (e.g. PCA, or nearest neighbor in the training set). Also, the proposed subdivision step seems harmful in Table 2 (and not helpful in Sup.Mat. Table 1), and it is unexpected to me: I think would be good including a comment in the text.
- Finally,  I understand that the focus of the paper is in a specific computer graphic domain, but while it is not the precise scope of the paper, I think would be interesting also to show the method limits when some of these priors elements are estimated for real-world cases (e.g. using pose regression techniques to infer the parameters). If this method can be applied also in real scenarios it would be really interesting.

2) Presentation:
I had some problems following the train of thought in some passages. I think it can be clearer by:
- Highlighting the contribution in the introduction, to clearly state the novelty of the work
- Provide a more structured taxonomy of the previous works for the reader. Also, I would suggest remarking the difference between the most important related papers and the proposed method.
- Grouping Data information in a single section (at the moment they are divided into Sections 4 and 6.1)
- Moving hyperparameters details to Supplementary Materials
- Moving the Methods with the Architecture (likely after the data section)

FINAL RATING
I think there is room to make the paper stronger; even if the paper proposes a simple method, with more evaluations, experiments, and analysis on different (and more challenging) examples it could find interest in the graphic community and future works.

---

> ### Author Response · Authors · 2020-11-24
> **Thank you for your feedback**
>
> We thank the reviewer for valuable feedback on our work. We provide clarifications to specific comments and concerns below.
>
> **“I think the proposed methodology is not particularly innovative, and it acts in a controlled scenario: the method is a CNN developed to work in a (virtual) setup that requires several prior knowledge: the view (also, it need to be trained for every different one), the pose parameters (making it also dependent on the animation system), the garment, the texture, and the light condition.”**
>
> For numerous cloth applications such as virtual try-on and AR/VR systems, the information listed above is known or can easily be inferred. For instance, any state-of-the-art body pose estimation network can be used to recover pose from RGB images, and the camera view, texture, and lighting condition can also be given or estimated. Using this knowledge as input to texture sliding does not mean our method can only be applied to a controlled scenario. Furthermore, our texture sliding method is not constrained to any specific garment - we use the T-shirt dataset to evaluate our method on available real world cloth data.
>
> **“Finally, the shown examples and experiments are limited to a few setups, without stressing the limits of the method. I am wondering how does the method performs training a different kind of garments with higher mobility (e.g. skirts) or subject to non-isometric deformations (e.g. zip clothes) or with irregular textures at higher frequencies (e.g. a shirt with a clear logo).”**
>
> Real world garments such as skirts or zipped clothes are currently unavailable, so we leave evaluation on such articles of clothing to future work. However, we note that predicting wrinkles/folds on a T-shirt is perhaps more challenging than complex outfits, where pre-baked wrinkles can be added to enhance realism rather than predicting all high frequency detail. In addition, we believe that the real textures used in our paper (checkerboard, stripes) best highlights cloth deformation.
>
> **“I am not surprised that the method improves the starting one, and I would suggest to compare it with others refinement baselines (e.g. PCA, or nearest neighbor in the training set). Also, the proposed subdivision step seems harmful in Table 2 (and not helpful in Sup.Mat. Table 1), and it is unexpected to me: I think would be good including a comment in the text.”**
>
> We consider the baseline for our method to be the inferred cloth from Jin et al. 2020, as the goal of the TSNN is to add high frequency detail to overly smooth network predictions. With regards to subdivision, we agree that the results are initially unexpected and updated Section E to comment on the results of Table 1 in the supplemental material.
>
> **“Finally, I understand that the focus of the paper is in a specific computer graphic domain, but while it is not the precise scope of the paper, I think would be interesting also to show the method limits when some of these priors elements are estimated for real-world cases (e.g. using pose regression techniques to infer the parameters). If this method can be applied also in real scenarios it would be really interesting.”**
>
> The cloth dataset we use for evaluation is a real world dataset, as the T-shirt and body were scanned from the real world and simulated for each pose. That said, we are interested in applying texture sliding to a wider range of garments and the application of cloth capture - these are avenues for future work.

---

### Official Review · AnonReviewer2 · 2020-10-28
**RECOVERING GEOMETRIC INFORMATION WITH LEARNED TEXTURE PERTURBATIONS**

**Rating:** 5
**Confidence:** 4

**Review:**

This work proposes an approach to encoding high frequency wrinkles into lower frequency texture coordinates, dubbed texture sliding.

At its core, is the idea of perturbing a predicted texture so that the rendered cloth mesh appears to more closely match the ground truth from a camera view point.
This is contrast to previous methods, which were attempting to make the network produce high-frequency components directly.
Once texture perturbations are recovered from at least two unique camera views, 3D geometry can then be reconstructed to recover high-frequency wrinkles.

Positive:

-The approach can be used to improve the quality of different architectures, although it requires laborious data pre-processing and to re-train a network for every specific architecture one wants to refine.

-Experimental results show the benefit of the proposed post-processing technique.


Negative:

-In the manuscript, it is mentioned that is necessary to train a separate network predicting texture perturbation for each camera. This makes the proposed pipeline cumbersome and computationally intensive. Why not conditioning prediction both on human pose and camera view point?

-The approach is claimed to be applicable to a general setting in the manuscript ("We focus on the specific task of adding highfrequency wrinkles to virtual clothing, noting that the idea of learning a low-frequency embedding may be generalized to other tasks”). However, both network architecture and the presented experiments are tailored to solve a very specific problem (getting wrinkles in pre-defined template t-shirt right). Unless other applications are presented, I find the authors claim to be misleading.


Clarifications:

-Are we assuming to have access to ground truth texture for the experiments? If so, this should be stated more clearly.

-Lifting slid textures to 3D vertex perturbations seems a non-trivial operation to me. Some math would have helped making the paper easier to read and more set-contained.

Justification of recommendation:

Although experimental results clearly show the benefit of the proposed method, I find several components of the pipeline to be to cumbersome to be used in practice (specifically: data re-generation per method, network re-training per method, network training per camera, relying on ground texture). Moreover, the proposed method feels more like a work-around to make things look better with clothing prediction, rather than an attempt to trying to solve the actual problem, i.e. learning high frequency information with neural networks.

Post author response:
After having carefully read the author's response and additional reviews, I confirm my original recommendation.

---

> ### Author Response · Authors · 2020-11-24
> **Thank you for your feedback**
>
> We thank the reviewer for valuable feedback on our work. We provide clarifications to specific comments and concerns below.
>
> **“-In the manuscript, it is mentioned that is necessary to train a separate network predicting texture perturbation for each camera. This makes the proposed pipeline cumbersome and computationally intensive. Why not conditioning prediction both on human pose and camera view point?”**
>
> Our texture sliding method is conditioned on the camera view point, and the input to the network is human pose. Thus, if only a single camera view is of interest (or two for AR/VR applications), only a single TSNN network needs to be trained.
>
> **“-The approach is claimed to be applicable to a general setting in the manuscript ("We focus on the specific task of adding highfrequency wrinkles to virtual clothing, noting that the idea of learning a low-frequency embedding may be generalized to other tasks”). However, both network architecture and the presented experiments are tailored to solve a very specific problem (getting wrinkles in pre-defined template t-shirt right). Unless other applications are presented, I find the authors claim to be misleading.”**
>
> As we state in the paper, “the idea of learning a low-frequency embedding may be generalized to other tasks.” Thus, we would like to clarify that the goal of this paper is indeed to address the problem of adding high frequency wrinkles/folds to virtual cloth. However, we note that our general approach of using a low frequency space (e.g. texture coordinates) to infer high frequency details (3D wrinkles/folds) may be applicable to other applications.
>
> **“-Are we assuming to have access to ground truth texture for the experiments? If so, this should be stated more clearly.”**
>
> Thanks for the suggestion - we have updated our discussion of the dataset in Section 5. The texture coordinates assigned to the ground truth cloth meshes was available to us for the experiments - however, we would like to emphasize that the “ground truth” data used for training the TSNN perturbs the texture coordinates of the inferred cloth (e.g. Jin et al. 2020) such that the cloth resembles the ground truth from a given camera view. Our dataset generation process is further explained in Section 4.
>
> **“-Lifting slid textures to 3D vertex perturbations seems a non-trivial operation to me. Some math would have helped making the paper easier to read and more set-contained.”**
>
> Please see Section 6.4 on how the output of the TSNN is used for multiview cloth reconstruction. We explain our algorithm in the paper; the reader can refer to our code for implementation details.

---

### Official Review · AnonReviewer4 · 2020-10-28
**It seems that this work needs substantial revisions by providing more details in a concise manner. Experimental results lack thorough comparative study with recent methods.**

**Rating:** 3
**Confidence:** 2

**Review:**

This paper presents a general approach to embed high frequency information into low-frequency data with a particular focus on improving the performance of virtual clothing. To address over-smoothing issues in the predicted meshes, authors proposed the texture sliding method that changes texture coordinates on each camera through the deep networks. The texture sliding neural network (TSNN) is trained using the ground truth offset computed for each camera and pose.

* Pros
1) The proposed TSNN leads to the performance gain in the virtual clothing

* Cons
1) The current manuscript requires major revisions by addressing the following concerns.
- It is hard to understand what to convey in Section 3.1. Especially, the following sentence just states the problem without how to address it.
‘This assumes linearity, which is only valid when the triangles are small enough to capture the inherent nonlinearities in a piecewise linear sense; moreover, folds and wrinkles can create significant nonlinearity.’
- It is difficult to follow what authors intend to convey in Figure 2.
- Two methods in Section 3.2 are hard to interpret.
- It is needed to mark both camera view and pose in Figure 4.
- Algorithm stating the overall procedure of the proposed method would improve the readability.
- What is 'UV space' in Section 5?
- How did you estimate the joint angles theta_k?
- What data did you use for training/validating/testing the networks?

2) Comparative study seems to be insufficient. It seems that a more thorough performance analysis is needed.

---

> ### Author Response · Authors · 2020-11-24
> **Thank you for your feedback**
>
> We thank the reviewer for valuable feedback on our work. We provide clarifications to specific comments and concerns below.
>
> **“It is hard to understand what to convey in Section 3.1. Especially, the following sentence just states the problem without how to address it. ‘This assumes linearity, which is only valid when the triangles are small enough to capture the inherent nonlinearities in a piecewise linear sense; moreover, folds and wrinkles can create significant nonlinearity.’”**
>
> Section 3.1 explains how taking a per-vertex, rather than per-point, approach to texture sliding can fail to capture nonlinearities in the texture when the triangles in a mesh are too large. This is because barycentric interpolation assumes linearity while folds and wrinkles are often highly nonlinear in nature. Figure 2 illustrates this effect.
>
> **“It is difficult to follow what authors intend to convey in Figure 2.“**
>
> Building on the discussion above, Figure 2 visualizes the errors caused by the linearity assumption of barycentric interpolation. The green triangle illustrates how the cloth texture would be barycentrically interpolated from the three triangle vertices *if the cloth mesh had such a large triangle*, but the yellow outline illustrates the region of the texture image that the triangle should encompass from the original cloth mesh.
>
> **“Two methods in Section 3.2 are hard to interpret.“**
>
> In Section 3.2 we do not present two methods, but rather consider the silhouette differences between the ground truth and inferred mesh when they are viewed from a specific camera view.
>
> **“It is needed to mark both camera view and pose in Figure 4.“**
>
> Figure 4 is a graphical illustration of our ray intersection method for texture sliding. It is not meant to depict ground truth data, thus we do not label the specific camera view and pose.
>
> **“Algorithm stating the overall procedure of the proposed method would improve the readability.“**
>
> Thanks for the suggestion. We describe the algorithm for our texture sliding data generation method in Section 4.1, and the reader can refer to our code for further details (which we can release).
>
> **“What is 'UV space' in Section 5?“**
>
> UV space refers to the process of UV mapping, i.e. the process of projecting a 2D texture image to a 3D mesh by assigning 2D texture coordinates in UV space.
>
> **“How did you estimate the joint angles theta_k?“**
>
> The joint angles are the input to the TSNN network, and are thus not estimated.
>
> **“What data did you use for training/validating/testing the networks?“**
>
> Please refer to Section 5 for details on the data we use, beginning with “The inferred cloth data we chose to correct are predictions of the T-shirt meshes from Jin et al. 2020…”
>
> **“Comparative study seems to be insufficient. It seems that a more thorough performance analysis is needed.“**
>
> We compare the TSNN (with and without subdivision) to Jin et al. 2020, which is a current state-of-the-art approach to cloth shape prediction. Our qualitative and quantitative results demonstrate that texture sliding effectively models material behavior and adds high frequency wrinkles/folds to overly smooth cloth predictions.

---

### Official Review · AnonReviewer3 · 2020-10-29
**Interesting idea, but the experiment is not good enough with many limitations**

**Rating:** 4
**Confidence:** 4

**Review:**

**Paper Summary**
The paper proposed a method to learn the geometric details of clothing mesh via perturbing texture coordinates. The paper first trains multiple networks (each network is for one specific camera view) to predict the texture coordinates in the corresponding camera view, then recover 3D shapes from multiple camera views.

**Strength**
1. Using texture sliding as a proxy to learn 3D geometry is an interesting idea
2. The paper also spends many efforts on designing texture sliding methods for occlusion boundary, which is hard for geometry processing.

**Weakness**
1. The paper claims to prevent overfitting induced from regularization and claim the proposed strategy can recover high-frequency details via procedurally embedded low-frequency data, however, the proposed method is specifically designed for 3D meshes (texture perturbation), it's not clear how does the method work with other machine learning representations, e.g. 2D image. At least, from my point view, this is not a trivial extension, and the paper didn't show the experiments beyond the 3D clothing mesh. I would suggest the authors turn down the claims in the paper.

2. The experiments on 3D reconstruction is not well enough. The paper argues to learn the geometric details, thus a careful analysis of the reconstructed 3D geometry (especially emphasizing on the geometric details) would be more convincing. However, the paper only presented one example of the final result (Fig.12/13), it would be better to: a) show more qualitative results, b) show quantitative comparisons with more baselines, e.g. a network that directly predicts the vertex offset in the clothing mesh. c). the quantitative analysis on the geometric details, how good the method can capture the details comparing with the baselines.

3. The scope of the paper is narrow. It's not trivial to extend the pipeline in the paper for a general machine learning representation. Also, in the paper, recovering 3D shapes requires post-processing and smoothing to make the shape look good.

**Overall**
Overall, Even though the idea of texture sliding is interesting, the experiments in this paper didn't demonstrate the clear improvements in recovering 3D geometric details. so I vote for a reject initially.

---

> ### Author Response · Authors · 2020-11-24
> **Thank you for your feedback**
>
> We thank the reviewer for valuable feedback on our work. We provide clarifications to specific comments and concerns below.
>
> **“The paper claims to prevent overfitting induced from regularization and claim the proposed strategy can recover high-frequency details via procedurally embedded low-frequency data, however, the proposed method is specifically designed for 3D meshes (texture perturbation), it's not clear how does the method work with other machine learning representations, e.g. 2D image. At least, from my point view, this is not a trivial extension, and the paper didn't show the experiments beyond the 3D clothing mesh. I would suggest the authors turn down the claims in the paper.”**
>
> We would like to clarify that the texture sliding method we propose is indeed designed for 3D meshes. However, our method advocates for a more general machine learning strategy of recovering high frequency detail via low-frequency embeddings.
>
> **“The experiments on 3D reconstruction is not well enough. The paper argues to learn the geometric details, thus a careful analysis of the reconstructed 3D geometry (especially emphasizing on the geometric details) would be more convincing. However, the paper only presented one example of the final result (Fig.12/13), it would be better to: a) show more qualitative results, b) show quantitative comparisons with more baselines, e.g. a network that directly predicts the vertex offset in the clothing mesh. c). the quantitative analysis on the geometric details, how good the method can capture the details comparing with the baselines.”**
>
> (a) Please see Appendix B in our supplemental material for more examples of reconstructed 3D geometry. (b) We compare the TSNN (with and without subdivision) to Jin et al. 2020, which is the only baseline for cloth shape inference with code and data made available to us. We note that SMPL-based methods require that the body is parameterized by the SMPL PCA basis, which is not the case with the body we use. (c) A quantitative analysis of how well our method captures geometric details is encapsulated by the sqrt MSE results we report. In addition, we visualized the per-vertex prediction errors in Figure 9c and provide more examples in the supplemental material (Figure 10c).
>
> **“The scope of the paper is narrow. It's not trivial to extend the pipeline in the paper for a general machine learning representation. Also, in the paper, recovering 3D shapes requires post-processing and smoothing to make the shape look good.”**
>
> Please see above for clarification on the scope of our paper. Adding high frequency wrinkles/folds to virtual cloth is an established area of work (discussed under “Wrinkles and Folds” in Section 2). While not required, the physical postprocess we use (Geng et al. 2020) further smooths the output of the TSNN and demonstrates that applying the postprocess to the TSNN results in greater wrinkling detail than when applied to Jin et al. 2020. See Figure 2 in the supplemental material.

---

### Author Response · Authors · 2020-11-24
**Submission Update**

We thank all the reviewers for their constructive comments. In this paper, our contributions focus on the specific application of predicting cloth geometry from 3D pose. Since the 3D cloth mesh can be rendered from any camera view, one only needs to train two TSNNs to reconstruct the ground truth cloth using triangulation. When one is only interested in the rendered cloth appearance, e.g. in 2D image space, there is also no need to train a different network for many different views as a small number of cameras can be strategically placed and interpolated from (as shown in Figure 11). Thus, the TSNN is both an efficient and generalizable approach to adding high-frequency details to overly smooth 3D geometry.

In accordance with reviewer feedback, we have made the following changes to our submission:
* We updated our discussion of the dataset in Section 5 to mention ground truth cloth mesh texture coordinates (R2).
* We added commentary in Section E of our supplemental material to address the results of subdividing the cloth meshes (R1).

---

### Decision · Program_Chairs · 2021-01-07
**Final Decision**

**Decision:**

Reject

**Comment:**

The 4 reviewers all had a consistent view of this paper:  concern that the scope of the work was overstated (paper claims, without evidence, to apply in more generality than the 1 example scenario shown); concern about the difficulty of implementing this approach (1 TSNN required for each rendered viewpoint); and lack of examples showing how the method performs under more challenging scenarios.

The AC encourages the authors to revise the work in response to the reviews.  That would involve additional experimentation and examples, and some attention to revising the manuscript.   After two of the reviewers complained of lack of clarity in the algorithm description, the authors replied, "We explain our algorithm in the paper; the reader can refer to our code for implementation details."  I hope the authors can be more responsive to the readers' concerns than that in their revisions.